# Unveiling Transformer Perception by Exploring Input Manifolds

## Abstract

This paper introduces a general method for the exploration of equivalence classes in the input space of Transformer models. The proposed approach is based on sound mathematical theory which describes the internal layers of a Transformer architecture as sequential deformations of the input manifold. Using eigendecomposition of the pullback of the distance metric defined on the output space through the Jacobian of the model, we are able to reconstruct equivalence classes in the input space and navigate across them. We illustrate how this method can be used as a powerful tool for investigating how a Transformer sees the input space, facilitating local and task-agnostic explainability in Computer Vision and Natural Language Processing tasks.

## 1 Introduction

In this paper, we propose a method for exploring the input space of Transformer models by identifying equivalence classes with respect to their predictions. We define an *equivalence class* of a Transformer model as the set of vectors in the embedding space whose outcomes under the Transformer process are the same. The study of the input manifold on which the inverse image of models lies provides insights for both explainability and sensitivity analyses. Existing methods aiming at the exploration of the input space of Deep Neural Networks and Transformers either rely on perturbations of input data using heuristic or gradient-based criteria [16, 22, 17, 14], or they analyze specific properties of the embedding space [5].

Our approach is based on sound mathematical theory which describes the internal layers of a Transformer architecture as sequential deformations of the input manifold. Using eigendecomposition of the pullback of the distance metric defined on the output space through the Jacobian of the model, we are able to reconstruct equivalence classes in the input space and navigate across them. In the XAI scenario, our framework can facilitate local and task-agnostic explainability methods applicable to Computer Vision (CV) and Natural Language Processing (NLP) tasks, among others.

In Section 2, we summarise the preliminaries of the mathematical foundations of our approach. In Section 3, we present our method for the exploration of equivalence classes in the input of the Transformer models. In Section 4, we perform a preliminary investigation of some applicability options of our method on textual and visual data. In Section 5, we discuss the relevant literature about embedding space exploration and feature importance. Finally, in Section 6, we give our concluding remarks[1].

---

[1]The code to reproduce our experiments can be found in the Supplementary Materials.

## 2  Preliminaries

In this Section, we provide the theoretical foundation of the proposed approach, namely the Geometric Deep Learning framework based on Riemannian Geometry [2].

A neural network is considered as a sequence of maps, the layers of the network, between manifolds, and the latter are the spaces where the input and the outputs of the layers belong to.

**Definition 1** (Neural Network). *A neural network is a sequence of $\mathcal{C}^1$ maps $\Lambda_i$ between manifolds of the form:*

$$M_0 \xrightarrow{\Lambda_1} M_1 \xrightarrow{\Lambda_2} M_2 \xrightarrow{\Lambda_4} \cdots \xrightarrow{\Lambda_{n-1}} M_{n-1} \xrightarrow{\Lambda_n} M_n \tag{1}$$

*We call $M_0$ the input manifold and $M_n$ the output manifold. All the other manifolds of the sequence are called representation manifolds. The maps $\Lambda_i$ are the layers of the neural network. We denote with $\mathcal{N}_{(i)} = \Lambda_n \circ \cdots \circ \Lambda_i : M_i \to M_n$ the mapping from the $i$-th representation layer to the output layer.*

As an example, consider a shallow network with just one layer, the composition of a linear operator $A \cdot + b$ with a sigmoid function $\sigma$, where $A \in \mathbb{R}^{m \times n}$ and $b \in \mathbb{R}^m$: then, the input manifold $M_0$ and the output manifold $M_1$ shall be $\mathbb{R}^n$ and $\mathbb{R}^m$, respectively, and the map $\Lambda_1(\cdot) = \sigma(A \cdot + b)$. We generalize this observation into the following definition.

**Definition 2** (Smooth layer). *A map $\Lambda_i : M_{i-1} \to M_i$ is called a smooth layer if it is the restriction to $M_{i-1}$ of a function $\overline{\Lambda}^{(i)}(x) : \mathbb{R}^{d_{i-1}} \to \mathbb{R}^{d_i}$ of the form*

$$\overline{\Lambda}^{(i)}_\alpha(x) = F^{(i)}_\alpha \left( \sum_\beta A^{(i)}_{\alpha\beta} x_\beta + b^{(i)}_\alpha \right) \tag{2}$$

*for $i = 1, \cdots, n$, $x \in \mathbb{R}^{d_i}$, $b^{(i)} \in \mathbb{R}^{d_i}$ and $A^{(i)} \in \mathbb{R}^{d_i \times d_{i-1}}$, with $F^{(i)} : \mathbb{R}^{d_i} \to \mathbb{R}^{d_i}$ a diffeomorphism.*

**Remark 1.** *Transformers implicitly apply for this framework, since their modules are smooth functions, such as fully connected layers, GeLU and sigmoid activations.*

Our aim is to transport the geometric information on the data lying in the output manifold to the input manifold: this allows us to obtain insight on how the network "sees" the input space, how it manipulates it for reaching its final conclusion. For fulfilling this objective, we need several tools from differential geometry. The first key ingredient is the notion of singular Riemannian metric, which has the intuitive meaning of a degenerate scalar product which changes point to point.

**Definition 3** (Singular Riemannian metric). *Let $M = \mathbb{R}^n$ or an open subset of $\mathbb{R}^n$. A singular Riemannian metric $g$ over $M$ is a map $g : M \to Bil(\mathbb{R}^n \times \mathbb{R}^n)$ that associates to each point $p$ a positive semidefinite symmetric bilinear form $g_p : \mathbb{R}^n \times \mathbb{R}^n \to \mathbb{R}$ in a smooth way.*

Without loss of generality, we can assume the following hypotheses on the sequence (1): *i)* The manifolds $M_i$ are open and path-connected sets of dimension $\dim M_i = d_i$. *ii)* The maps $\Lambda_i$ are $\mathcal{C}^1$ submersions. *iii)* $\Lambda_i(M_{i-1}) = M_i$ for every $i = 1, \cdots, n$. *iv)* The manifold $M_n$ is equipped with the structure of Riemannian manifold, with metric $g^{(n)}$. Definition 3 naturally leads to the definition of the pseudolength and of energy of a curve.

**Definition 4** (Pseudolength and energy of a curve). *Let $\gamma : [a, b] \to \mathbb{R}^n$ a curve defined on the interval $[a, b] \subset \mathbb{R}$ and $\|v\|_p = \sqrt{g_p(v, v)}$ the pseudo–norm induced by the pseudo–metric $g_p$ at point $p$. Then the pseudolength of $\gamma$ and its energy are defined as*

$$Pl(\gamma) = \int_a^b \|\dot{\gamma}(s)\|_{\gamma(s)} ds = \int_a^b \sqrt{g_{\gamma(s)}(\dot{\gamma}(s), \dot{\gamma}(s))} ds, \qquad E(\gamma) = \int_a^b \|\dot{\gamma}(s)\|^2_{\gamma(s)} ds \tag{3}$$

The notion of pseudolength leads naturally to define the distance between two points.

**Definition 5** (Pseudodistance). *Let $x, y \in M = \mathbb{R}^n$. The pseudodistance between $x$ and $y$ is then*

$$Pd(x, y) = \inf\{Pl(\gamma) \mid \gamma : [0, 1] \to M, \gamma \in \mathcal{C}^1([0, 1]), \gamma(0) = x, \gamma(1) = y\}. \tag{4}$$

One can observe that endowing the space $\mathbb{R}^n$ with a singular Riemannian metric leads to have non trivial curves whose length is zero. A straightforward consequence is that there are distinct points whose pseudodistance is therefore zero: a natural equivalence relation arises, *i.e.* $x \sim y \Leftrightarrow Pd(x, y) = 0$, obtaining thus a metric space $(\mathbb{R}^n / \sim, Pd)$.

The second crucial tool is the notion of pullback of a function. Let $f$ be a function from $\mathbb{R}^p$ to $\mathbb{R}^q$, and fix the coordinate systems $x = (x_1, \dots, x_p)$ and $y = (y_1, \dots, y_q)$ on $\mathbb{R}^p$ and on $\mathbb{R}^q$, respectively. Moreover, we endow $\mathbb{R}^q$ with the standard Euclidean metric $g$, whose associated matrix is the identity. The space $\mathbb{R}^p$ can be equipped with the pullback metric $f^*g$ whose representation matrix reads as

$$(f^*g)_{ij} = \sum_{h,k=1}^{q} \left(\frac{\partial f_h}{\partial x_i}\right) g_{hk} \left(\frac{\partial f_k}{\partial x_j}\right). \tag{5}$$

The sequence (1) shows that a neural network can be considered simply as a function, a composition of maps: hence, taking $f = \Lambda_n \circ \Lambda_{n-1} \circ \cdots \circ \Lambda_1$ and supposing that $M_0 = \mathbb{R}^p, M_n = \mathbb{R}^q$, the generalization of (5) applied to (1) provides with the pullback of a generic neural network.

Hereafter, we consider in (1) the case $M_n = \mathbb{R}^q$, equipped with the trivial metric $g^{(n)} = I_q$, *i.e.*, the identity. Each manifold $M_i$ of the sequence (1) is equipped with a Riemannian singular metric, denoted with $g^{(i)}$, obtained via the pullback of $\mathcal{N}_{(i)}$. The pseudolength of a curve $\gamma$ on the $i$-th manifold, namely $Pl_i(\gamma)$, is computed via the relative metric $g^{(i)}$ via (3).

## 2.1   General results

We depict hereafter the theoretical bases of our approach. We denote with $\mathcal{N}_i$ the submap $\Lambda_i \circ \cdots \circ \Lambda_n :$ $M_i \to M_n$, and with $\mathcal{N} \equiv \mathcal{N}_0$ the map describing the action of the complete network. The starting point is to consider the pair $(M_i, Pd_i)$: this is a pseudometric space, which can be turned into a full-fledged metric space $M_i / \sim_i$ by the metric identification $x \sim_i y \Leftrightarrow Pd_i(x, y) = 0$. The first result states that the length of a curve on the $i$-th manifold is preserved among the mapping on the subsequent manifolds.

**Proposition 1.** *Let $\gamma : [0, 1] \to M_i$ be a piecewise $\mathcal{C}^1$ curve. Let $k \in \{i, i+1, \cdots, n\}$ and consider the curve $\gamma_k = \Lambda_k \circ \cdots \circ \Lambda_i \circ \gamma$ on $M_k$. Then $Pl_i(\gamma) = Pl_k(\gamma_k)$.*

In particular this is true when $k = n$, *i.e.*, the length of a curve is preserved in the last manifold. This result leads naturally to claim that if two points are in the same class of equivalence, then they are mapped into the same point under the action of the neural network.

**Proposition 2.** *If two points $p, q \in M_i$ are in the same class of equivalence, then $\mathcal{N}_i(p) = \mathcal{N}_i(q)$.*

The next step is to prove that the sets $M_i / \sim_i$ are actually smooth manifolds: to this aim, we introduce another equivalence relation: $x \sim_{\mathcal{N}_i} y$ if and only if there exists a piecewise $\gamma : [0, 1] \to M_i$ such that $\gamma(0) = x, \gamma(1) = y$ and $\mathcal{N}_i \circ \gamma(s) = \mathcal{N}_i(x)\ \forall s \in [0, 1]$. The introduction of this equivalence relation allows us to easily state the following proposition.

**Proposition 3.** *Let $x, y \in M_i$, then $x \sim_i y$ if and only if $x \sim_{\mathcal{N}_i} y$.*

The following corollary contains the natural consequences of the previous result; the second point of the claim below is the counterpart of Proposition 2.

**Corollary 1.** *Under the hypothesis of Proposition 3, one has that $M_i / \sim_i = M_i / \sim_{\mathcal{N}_{i+1}}$. Moreover, if two points $p, q \in M_i$ are connected by a $\mathcal{C}^1$ curve $\gamma : [0, 1] \to M_i$ satisfying $\mathcal{N}_i(p) = \mathcal{N}_i \circ \gamma(s)$ for every $s \in [0, 1]$, then they lie in the same class of equivalence.*

Making use of the Godement's criterion, we are now able to prove that the set $M_i / \sim_i$ is a smooth manifold, together with its dimension.

**Proposition 4.** $\dfrac{M_i}{\sim_i}$ *is a smooth manifold of dimension* $dim(\mathcal{N}(M_0))$.

This last achievement provides practical insights about the projection $\pi_i$ on the quotient space, that consists the building block of the algorithms used for recovering and exploring the equivalence classes of a neural network.

**Proposition 5.** $\pi_i : M_i \to M_i/\sim_i$ *is a smooth fiber bundle, with* $Ker(d\pi_i) = \mathcal{V}M_i$, *which is therefore an integrable distribution.* $\mathcal{V}M_i$ *is the vertical bundle of* $M_i$. *Every class of equivalence* $[p]$ *is a path-connected submanifold of* $M_i$ *and coincide with the fiber of the bundle over the point* $p \in M_i$.

# 3  Methodology

The results depicted in Section 2.1 provide powerful tools for investigating how a neural network sees the input space starting from a point $x$. In particular we point out the following remarks: i) If two points $x, y$ belonging to the input manifold $M_0$ are are such that $x \sim_0 y$, then $\mathcal{N}(x) = \mathcal{N}(y)$; ii) given a point $p \in M_n$, the counterimage $\mathcal{N}^{-1}(p)$ is a smooth manifold, whose connected components are classes of equivalences in $M_0$ with respect to $\sim_0$. A necessary condition for two points $x, y \in M_0$ to be in the same class of equivalence is that $\mathcal{N}(x) = \mathcal{N}(y)$; iii) any class of equivalence $[x]$, $x \in M_0$, is a maximal integral submanifold of $\mathcal{V}M_0$. The above observations directly provide with a strategy to build up the equivalence class of an input point $x \in M_0$. Proposition 5 tells us that $\mathcal{V}M_0$ is an integrable distribution, with dimension equal to the dimension of the kernel of $g^{(0)}$: we can hence find $dim(Ker(g^{(0)}))$ vector fields which are a base for the tangent space of $M_0$. This means that we can compute the eigenvalue decomposition of $g_x^{(0)}$ and consider the $L$ linearly independent eigenvectors, namely $\{v_l\}_{l=1,\dots,L}$, associated to the null eigenvalue: these eigenvectors depend *smoothly* on the point, a fact that is not trivial when the matrix associated to the metric depends on several parameters [15]. We can build then all the null curves by randomly selecting one eigenvector $\tilde{v} \in \{v_l\}$ and then reconstruct the curve along the direction $\tilde{v}$ from the starting point $x$. From a practical point of view, one is led to solve the Cauchy problem, a first order differential equation, with $\dot{\gamma} = \tilde{v}$ and initial condition $\gamma(0) = x$.

## 3.1  Input Space Exploration

This whole procedure is coded in the Singular Metric Equivalence Class (SiMEC) and the Singular Metric Exploration (SiMExp) algorithms, whose general schemes are depicted in Algorithms 1 and 2. SiMEC reconstructs the class of equivalence of the input via the exploration of the input space by randomly selecting one of the eigenvectors related to the zero eigenvalue. On the opposite, in SiMExp, in order to move from a class of equivalence to another we consider the eigenvectors relative to the nonzero eigenvalues. This requires the slight difference in lines 5 to 7 between Algorithm 1 and Algorithm 2.

---

**Algorithm 1** The Singular Metric Equivalence Class (SiMEC) algorithm.

1: Set the network $\mathcal{N}$; choose the maximum number of iterations. Choose the input $p_0$.
2: **for** $k = 0, 1, \dots, K - 1$ **do**
3:     Compute $g_{\mathcal{N}(p_k)}^n$
4:     Compute the pullback metric $g_{p_k}^0$
5:     Diagonalize $g_{p_k}^0$ and find the eigenvectors $\{v_l\}_l$ associated to the zero eigenvalue
6:     Randomly select $\tilde{v} \in \{v_l\}_l$
7:     $\delta = 1/\sqrt{\max(\text{eigenvalues of } g_{p_k}^0)}$
8:     $p_{k+1} \leftarrow p_k + \delta \tilde{v}$
9: **end for**
10: Optionally: store $\{p_k\}_{k=0,\dots,K}$ for optimizing future computations
11: Project $p_k$ to the nearest feasible region

**Algorithm 2** The Singular Metric Exploration (SiMExp) algorithm.

1: Set the network $\mathcal{N}$; choose the maximum number of iterations. Choose the input $p_0$.
2: **for** $k = 0, 1, \dots, K - 1$ **do**
3:     Compute $g_{\mathcal{N}(p_k)}^n$
4:     Compute the pullback metric $g_{p_k}^0$
5:     Diagonalize $g_{p_k}^0$ and find the eigenvectors $\{w_l\}_l$ associated to the non-zero eigenvalue
6:     Randomly select $\tilde{w} \in \{w_l\}_l$
7:     $\delta = 2/\sqrt{\max(\text{eigenvalues of } g_{p_k}^0)}$
8:     $p_{k+1} \leftarrow p_k + \delta \tilde{w}$
9: **end for**
10: Optionally: store $\{p_k\}_{k=0,\dots,K}$ for optimizing future computations
11: Project $p_k$ to the nearest feasible region

---

There are some remarks to point out. From a numerical point of view, the diagonalization of the pullback may lead to have even negative eigenvalues: hence one may use the notion of energy of a curve, related to the pseudolength. The update rule for the new point (line 8) amounts to solve the differential problem via the Euler method: for a reliable solution, we suggest to choose a small step-length $\delta$. On the other hand, if the value of $\delta$ is too small more iterations are needed to move

151 away from the starting point sensibly. Therefore there is a trade-off between the reliability of the
152 solution and the exploration pace. The proof of the well-posedness theorem for Cauchy problems,
153 cf. [18, Theorem 2.1], yields some insights, suggesting to set $\delta$ equal to the inverse of the Lipschitz
154 constant of the map $\mathcal{N}$ – which in practice we can estimate with the inverse of the square root of the
155 largest eigenvalue $\lambda_M$ of the pullback metric $g_{p_k}^0$. This is our default choice for Algorithm 1. We also
156 note that Algorithm 1 is more sensitive to the choice of the parameter $\delta$ compared to Algorithm 2.
157 To build points in the same equivalence class Algorithm 1 needs to follow a null curve closely with
158 as little approximations as possible, namely with a small $\delta$. In contrast Algorithm 2, whose goal is
159 to change the equivalence class from one iteration to the next, does not have the same problem and
160 larger $\delta$ are allowed. Out default choice is therefore to set $\delta = 2\lambda_M^{-1/2}$ for Algorithm 2. As for the
161 computational complexity of the two algorithms, the most demanding step is the computation of the
162 eigenvalues and eigenvectors, which is $O(n^3)$, with $n$ the dimension of the square matrix $g_{p_k}^0$ [20].
163 Since all the other operations are either $O(n)$ or $O(n^2)$, we conclude that the complexity of both
164 Algorithms 1 and 2 is $O(n^3)$.

## 3.2 Interpretability

166 Algorithms 1 and 2 allow for the exploration of the equivalence classes in the input space of a Trans-
167 former model. However, the points explored by these algorithms may not be directly interpretable
168 by a human perspective. For instance, an image or a piece of text may need to be decoded to be
169 "readable" by a human observer. Furthermore, we present an interpretation of the eigenvalues of the
170 pullback metric which allows us to define a feature importance metric. We present two interpretability
171 methods for Transformers based on input space exploration. Both methods are then demonstrated on
172 a Vision Transformer (ViT) trained for digit classification [8], and two BERT models, one trained for
173 hate speech classification and the other trained for MLM [7, 19].

174

---

**Algorithm 3** Feature Importance Analysis Using Pull-back Metric $g_{x^e}^0$

1: **Inputs:**
2:      Transformer model $T$ with: Tokenizer $t_T$, Embedding layer $e_T$, Intermediate layers $l_T$
3:      Input data $x$
4: Tokenize input $x$ to obtain tokens $x^t = t_T(x)$
5: Compute embeddings $x^e = e_T(x^t)$
6: Compute intermediate representations $g_{l_T(x^e)}^n$
7: Calculate the pullback metric $g_{x^e}^0$
8: Diagonalize $g_{x^e}^0$ to extract eigenvalues
9: Identify the maximum eigenvalue for each embedding, indicating its importance
10: **Output:** Heatmap of embedding importance based on the eigenvalues

---

**Algorithm 4** Exploration of Embedding Space in Transformers

1: **Inputs:**
2:      Transformer model $T$ with: Tokenizer $t_T$, Embedding layer $e_T$, Intermediate layers $l_T$
3:      Input data $x$ (image or text)
4: Retrieve segments $x^t = t_T(x)$.
5: Choose segments $P = \{p|p \in x^t\}$ for updates; keep others unchanged.
6: Compute embeddings $x^e = e_T(x^t)$.
7: Apply SiMEC or SiMExp on $x^e$, updating embeddings for segments in $P$.
8: **Outputs:** Modified input embedding, one for each SiMEC/SiMExp iteration.

---

175 **Feature importance.**   Consider a Transformer model $T$ whose architecture includes a tokenizer $t_T$
176 (or patcher for images) that segments the input so that each segment can be converted into a continuous
177 representation by an embedding layer $e_T$. This results in a matrix of dimensions $n_s \times h$, where $n_s$
178 represents the number of segments, and $h$ denotes the hidden size of the model's embeddings. The
179 eigenvalues of the pullback metric can be used to deduce the importance of each embedding and, by
180 extension, the significance of the segments they represent, with respect to the final prediction. The
181 process for determining the importance of textual tokens or image patches is outlined in Algorithm 3.
182 The appearance of the resulting heatmaps varies according to the type of input used. An example
183 of experiments with ViT on the MNIST dataset [12] is shown in Figure 1 that depicts heatmaps for
184 two MNIST instances. Figure 2, on the left, illustrates two experiment using Algorithm 3 on both a
185 BERT model for hate speech detection and a BERT model for MLM.

186 **Interpretation of input space exploration.**   Using SiMEC and SiMExp to explore the embedding
187 space reveals how Transformer models perceive equivalence among different data points. Specifically,
188 these methodologies facilitate the sequential acquisition of embedding matrices $p_0 \ldots p_K$ at each
189 iteration, as detailed in Algorithms 1 and 2. Algorithm 4 implements a practical application of the

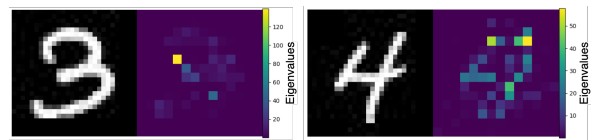

Figure 1: Example output from Algorithm 3 applied to digit classification. These two instances are predicted as 3 (left) and 4 (right). The brightness of the color indicates the eigenvalue's magnitude. The brighter the color, the more sensitive the patch. This indicates that changes in the values of these sensitive patches are likely to have a greater impact on the prediction probabilities. Each patch in the heatmap corresponds to a $2 \times 2$ square pixel.

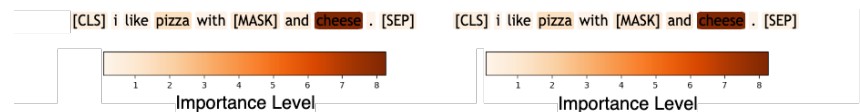

Figure 2: Example outputs from Algorithm 3. The darker the color, the higher the token's eigenvalue. Left: The sentence analysed is classified as "offensive" by the BERT for hate speech detection, with significant contributions from tokens [CLS], politicians, corrupt, and ##eit (part of the word *deceitful*). Right: Example instance processed by a BERT model for masked language modeling. [MASK] is predicted as "ham", with the most influential tokens being pizza and cheese.

SiMEC/SiMExp approach with Transformer models. A key feature of this method is its ability to selectively update specific tokens (for text inputs) or patches (for image inputs) during each iteration. This selective updating allows us to explore targeted modifications that prompt the model to either categorize different inputs as the same class or recognize them as distinct. Unlike traditional approaches where modifications are predetermined, this method lets the model itself guide us to understand which data points belong to specific equivalence classes. To interpret embeddings resulted from the exploration process, they must be mapped back into a human-understandable form, such as text or images. The interpretation of an embedding vector depends on the operations performed by the Transformer's embedding module $e_T$. If $e_T$ consists only of invertible operations, it is feasible to construct a layer that performs the inverse operation relative to $e_T$. The output can then be visualized and directly interpreted by humans, allowing for a comparison with the original input to discern how differences in embeddings reflect differences in their representations (e.g., text, images). If the operations in $e_T$ are non-invertible, a trained decoder is required to reconstruct an interpretable output from each embedding matrix $p_0 \ldots p_K$. When using a BERT model, it is feasible to utilize layers that are specialized for the masked language modeling (MLM) task to map input embeddings back to tokens. This approach is effective whether the BERT model in question is specifically designed for MLM or for sentence classification. In the case of sentence classification models, it is necessary to select a corresponding MLM BERT model that shares the same internal architecture, including the number of layers and embedding size.

Algorithm 5 depicts the process of interpreting Algorithm 4 outputs for both ViT and BERT experiments. After initializing the decoder according to the model type, the embeddings $p_0 \ldots p_K$ need to be constrained to a feasible region. This region is defined by the distribution of embeddings derived from the original input instances. Next, the embeddings are decoded, and the selected segments for exploration are extracted. These segments are then used to replace the corresponding parts of the original input instance. Figure 3 depicts an example outcome of Algorithm 5 applied on a ViT exploration experiment. Given that the interpretation process includes both a capping step and a decoding step (lines 10 and 11 of Algorithm 5), it's important to note that there isn't a direct 1:1 correspondence between each iteration's update and the interpretation outcomes. Our primary focus is on exploring the input embedding space, rather than the input image or input sentence spaces. For further investigation, we provide a detailed discussion on considering interpretation outputs as alternative prompts in Section 4.

---

**Algorithm 5** Interpretation for Exploration results for ViT and BERT models.

---

1: **Inputs:**
2:     Transformer model $T$ with: Tokenizer $t_T$, Embedding layer $e_T$, Intermediate layers $l_T$
3:     Modified embeddings $p_0 \dots p_K$ resulted from Algorithm 4 applied on an input $x$
4:     $P = \{p | p \in x^t\}$ indices of updated segments
5: **If** $T$ **is** ViT:
6:     Initialize decoder $d$ with weights from $e_T$.
7: **If** $T$ **is** BERT:
8:     Initialize decoder with intermediate and final layers of a BERT for MLM task.
9: Compute embeddings distributions for original input data
10: Use the original embeddings distributions to cap $p_0 \dots p_K$
11: Decode modified embeddings $p_0 \dots p_K$ using $d$ to generate the corresponding images/sentences $X' = x'_0 \dots x'_K$.
12: **For each** $x' \in X'$: replace segments relative to indices $P$ in $x$ with those in $x'$.
13: **Outputs:**
14: Modified input images/sentences, one for each SiMEC/SiMExp iteration.

---

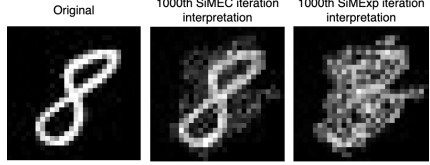

Figure 3: Example of SiMEC and SiMExp output interpretation for ViT digit classification. Left: Original MNIST image of an "8". Center: Interpretation of a $p_{1000}$ from a SiMEC experiment, where $p_{1000}$ is predicted as "8". Right: Interpretation of a $p_{1000}$ from a SiMExp experiment, where $p_{1000}$ is predicted as "4". All patches are subject to SiMEC and SiMExp updates.

# 4 Experiments

Experiments are conducted on textual and visual data. We aim to perform a preliminary investigation of 3 features of our approach: (i) how the class probability changes on the decoded output of SiMEC/SiMExp, (ii) what is the trade-off between the quantity and the quality of the output, and (iii) how our method can be used to extract feature importance-based explanations.

In the textual case, we experiment with hate speech classification datasets: we use HateXplain[2] [13], which provides a ground truth for feature importance, plus a sample of 100 hate speech sentences generated by prompting ChatGPT[3], which serve purposes (i) and (ii). In the visual case, we perform experiments on MNIST [12] dataset.

**Using interpretation outputs as alternative prompts**    An interesting investigation is to determine if our interpretation algorithm (Algorithm 5) can generate alternative prompts that stay in the same equivalence class as the original input data or move to a different one, based on SiMEC and SiMExp explorations. We test how the probability assigned to the original equivalence class by the Transformer model changes as the SiMEC and SiMExp algorithms explore the input embedding manifold.

For BERT experiments we generate prompts to inspect the probability distribution over the vocabulary for tokens updated by Algorithms 1 and 2. We decode the updated $p_0 \dots p_K$ using Algorithm 5, focusing on tokens updated through the iterations. For each of these decoded tokens, we extract the top-5 scores to obtain 5 alternative tokens to replace the original ones, creating 5 alternate prompts. We then extract the prediction $i^* = \arg\max_i y_i$ for the original sentence, which represents the output whose equivalence class we aim to explore. Finally, we classify the new prompts, obtaining the corresponding predictions $Y = \mathbf{y}^{(0)} \dots \mathbf{y}^{(K)}$, where each $\mathbf{y}^{(k)} \in \mathbb{R}^N$, $N$ being the number of prediction classes. We visualize the prediction trend for the $i^*$th value in every $\mathbf{y}^{(0)} \dots \mathbf{y}^{(K)}$ categorizing the images into two subsets: those that lead to a change in prediction $Y_c = \{\mathbf{y}^{(k)} \in Y \mid \arg\max_i y_i^{(k)} \neq i^*\}$ and those that don't $Y_s = \{\mathbf{y}_i \in Y \mid \arg\max_i y_i^{(k)} = i^*\}$.

---

[2]MIT License
[3]Used prompts are included in the Supplementary Materials.

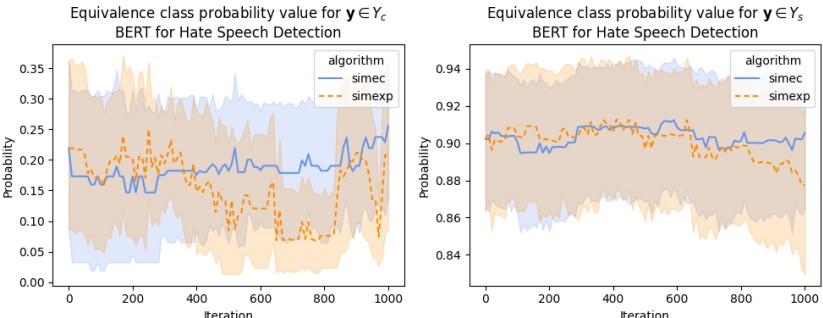

Figure 4: Analysis involving results SiMEC and SiMExp applied to BERT for hate speech detection. Left: Prediction values for $i^*$ for each $\mathbf{y} \in Y_c$. Right: Prediction values for $\mathbf{y} \in Y_s$.

Sentence classification experiments[4] involved 1000 iterations from both SiMEC and SiMExp, applied to a subset of 8 sentences from the ChatGPT hate speech dataset. The plot on the left side of Figure 4 illustrates that, as the original embeddings are increasingly modified, SiMExp tends to produce alternatives with lower prediction values for $i^*$ compared to SiMEC. Thus, even if predictions change in SiMEC experiments, the equivalence class prediction value remains approximately constant and higher than in SiMExp. Considering the plot on the right side of Figure 4, SiMExp identifies prompts that lower the prediction value for $i^*$. ViT and MLM experiments are detailed in the Supplementary Materials.

**Input space exploration** We measure the time required to explore the input space of a ViT with the SiMEC algorithm and compare it with a perturbation-based method. The perturbation-based method mimics a trial-and-error approach as it takes an input image and, at each iteration, perturbs it by a semi-random vector $\mathbf{v}_{t+1} = a_t \mathbf{v}_t + \eta \epsilon$, where $a_t = 1$ if $y_t = y_{t-1}$, $a_t = -1$ otherwise, $\epsilon$ is an orthogonal random vector from a standard normal distribution and $\eta$ is the step length. With the perturbation, we obtain a new image, then check whether the model yields the same label for the new image. The perturbation vector is re-initialized at random from a normal distribution 20% of the times to allow for exploration. We construct this method to have a direct comparison with ours in the absence of a consolidated literature about the task.

We train a ViT model having 4 layers and 4 heads per layer on the MNIST dataset[5]. The SiMEC algorithm is run for 1000 iterations, so that it can generate 1000 examples starting from a single image. In a sample of 100 images, the average time is approximately 339 seconds.[6] In the same time, the perturbation-based algorithm can produce up to 36000 images. However, we notice that the perturbation-based algorithm ends up producing monochrome (pixel color has zero variance) or totally noisy images, which provide little information about the behavior of the model. Excluding only the images with low color variance ($< 0.01$), we are left, on average, with 19 images (standard deviation 13.9). SiMEC, in contrast, doesn't present this behavior, as all 1000 images have high enough intensity variance and are thus useful for explainability purposes.

As BERT has many more parameters with respect to our ViT model, processing textual data takes longer. Specifically, in a sample of 16 sentences, the average time needed to run 1000 iterations on a sentence is 7089 seconds, taking into account both MLM and classification experiments.

**Feature importance-based explanations** We compare our method against Attention Rollout (AR) [1] and the Relevancy method proposed by Chefer et al. [6]. In the textual case, we provide a quantitative evaluation using the HateXplain dataset, which contains 20147 sentences (of which 1924 in the test set) annotated with *normal*, *offensive* and *hate speech* labels as well as the positions of words that support the label decision. We then measure the cosine similarity between the importance assigned by each method to each word in a sentence and the ground truth. Notice that, since the

---

[4]Model used: `huggingface.co/ctoraman/hate-speech-bert`

[5]Using Adam optimizer, the model achieved the highest validation accuracy (96.25%) in 20 epochs.

[6]All experiments are based on the current PyTorch implementation of the algorithms and run on a Ubuntu 20.04 machine endowed with one NVIDIA A100 GPU and CUDA 12.4.

dataset contains multiple annotations, the ground truth $y$ for each word $w$ is obtained as the average of the binary labels assigned by each annotator, and therefore $y(w) \in [0; 1]$. We also normalize all scores in $[0; 1]$ so to have them on the same scale. The average similarity achieved by our method is **0.707** (standard deviation $\sigma = 0.302$), against **0.7** ($\sigma = 0.315$) for Relevancy and **0.583** ($\sigma = 0.318$) for AR. This proves our method to be more effective in finding the most sensitive tokens for classification. We provide an example on image classification in the Supplementary Materials.

## 5  Related work

Our work relates to embedding space exploration literature, and has at least one collateral applications in the XAI domain, namely producing feature importance-based explanations.

**Embedding space exploration.**    Works dealing with embedding space exploration mostly focus on the study of specific properties of the embedding space of Transformers, especially in NLP. For instance, Cai et al. [5] challenge the idea that the embedding space is inherently anisotropic [10] discovering local isotropy, and find low-dimensional manifold structures in the embedding space of GPT and BERT. Biś et al. [3] argue that the anisotropy of the embedding space derives from embeddings shifting in common directions during training. In the field of CV, Vilas et al. [21] map internal representations of a ViT onto the output class manifold, enabling the early identification of class-related patches and the computation of saliency maps on the input image for each layer and head. Applying Singular Value Decomposition to the Jacobian matrix of a ViT, Salman et al. [17] treat the input space as the union of two subspaces: one in which image embedding doesn't change, and another one for which it changes. Except for the last one, all the aforementioned approaches rely on data samples. By studying the inverse image of the model, instead, we can do away with data samples.

**Feature importance-based explanations.**    Feature importance is a measure of the contribution of each data feature to a model prediction. In the context of Computer Vision and Natural Language Processing, it amounts to giving a weight to pixels (or patches of pixels) in an image and tokens in a piece of text, respectively. In recent years, much research has focused on Transformers in both CV and NLP. Most approaches are based on the attention mechanism of the Transformer architecture. Abnar and Zuidema [1] quantify the overall attention of the output on the input by computing a linear combination of layer attentions (Attention Rollout) or applying a maximum flow algorithm (Attention Flow). To overcome the limitations [4] of attention-based methods, Hao et al. [11] use the concept of *attribution*, which is obtained by multiplying attention matrices by the integrated gradient of the model with respect to them. Chefer et al. [6] propose the Relevancy metric to generalize attribution to bi-modal and encoder-decoder architectures. Other methods are perturbation-based, where perturbations of input data are used to record any change in the output and draw a saliency map on the input. In order to overcome the main issue with such methods, i.e. the generation of outlier inputs, Englebert et al. [9] apply perturbations after the position encoding of the patches. In contrast with these methods, ours does not need arbitrary perturbations of inputs, and considers all parameters of the model, not only the attention query and key matrices.

## 6  Conclusions

Our exploration of the Transformer architecture through a theoretical framework grounded in Riemannian Geometry led to the application of our two algorithms, SiMEC and SiMExp, for examining equivalence classes in the Transformers' input space. We demonstrated how the results of these exploration methods can be interpreted in a human-readable form and conducted preliminary investigations into their potential applications. Notably, our methods show promise for ranking feature importance and generating alternative prompts within the same or different equivalence classes.

Future research directions include expanding our experimental results and delving deeper into the potential of our framework for controlled input generation within an equivalence class. This application holds significant promise for enhancing the explainability of Transformer models' decisions and for addressing issues related to bias and hallucinations.

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
