# OpenReview forum: "Unveiling Transformer Perception by Exploring Input Manifolds"
_NeurIPS.cc/2024/Conference — Submitted to NeurIPS 2024_

### Official Review · Reviewer_KhQb · 2024-06-27

**Soundness:** 3
**Presentation:** 4
**Contribution:** 3
**Rating:** 4
**Confidence:** 2

**Summary:**

The authors propose a mathematically rigorous methodology based on Riemannian geometry for attributing network importance of tokens in a transformer models input space (e.g. image patches, or ~words in the textual domain). The proposed methodology—whilst based on sound theory—translates into an intuitive algorithm involving what appears to be a relatively inexpensive eigendecomposition. Experiments on 3 datasets across both the image and NLP domains explore how the features correlate with ground-truth inputs in the text domain, in addition to first steps towards exploring how the features affect the networks’ output logits.

**Strengths:**

- A major strength of the paper is the mathematically solid approach in attempting to identify regions of the **input** space that explain transformers’ model decisions. This is an important area of study: in contrast to many recent mechanistic interpretability methods finding latent network representations (that are intrinsically hard for humans to interpret by default), salient features in the pixel/text input space are much more readily interpreted by humans.
- Whilst I am unfamiliar with geometric deep learning, the authors do a fantastic job of presenting the technical content in a digestible manner without sacrificing depth or rigor.

**Weaknesses:**

# [W1] Feature importance comparisons

Feature importance-based explanations are motivated on [L303] as quantifying the contribution of features `"to a model prediction"`. More concretely, around [L180], the authors motivate the eigenvalues of the pullback metric found using their method as ultimately deducing the importance of each segment (e.g. image patch)  `“with respect to the the final prediction”`.

Consequently, a major weakness of the paper is how there is no comparison with related work for how well the proposed method’s identified important features alter the **output** logits (e.g. upon ablation).

I am slightly confused by why the authors did not adopt the established “perturbation test” experimental protocol in the baseline [1] against which they compared, to provide experimental evidence in favor of this. Currently, the only comparisons made around [L274] measure the features’ importance as they correlate to the *input’s* labels.

Concretely, the authors could, for example, ablate particular patches of MNIST and observe that the resulting performance drops correlate with the pullback metric’s eigenvalues. This would provide stronger evidence of the authors’ claims about the features affecting the networks’ output, and (crucially) ground the results in contrast to those achievable by existing methods.

# [W2] Limited experimental results & improvements

There is a lot of interesting theory here, but ultimately this is a paper with a concrete applied goal of feature attribution in transformer models. With such a new methodology with many technical details, I believe there is an extra burden of proof on the authors to demonstrate this somehow leads to additional insights / practical gains. As such, it is a relative weakness of the paper that so few experiments are performed to justify the methodology.

Beyond toy datasets, it would be interesting to see how the method performs on more complex ones (not necessarily larger ones), such as TinyImageNET. Here, we could visualize much more easily if the method helps identify salient features of animals’ body parts (for example) as being important features for classification. MNIST experiments alone in the image domain are hard to interpret given the similarity of all the input data.

Furthermore, the method provides an almost insignificant increase of just `0.07` cosine similarity (over the baseline in [1]), on just a single dataset (and with just two baselines—for example, how does GradCAM perform here?). This is not sufficient evidence to convince me as a reader that the proposed methodology should be adopted.

---

- [1]: Chefer, Hila et al. “Generic Attention-model Explainability for Interpreting Bi-Modal and Encoder-Decoder Transformers.” 2021 IEEE/CVF International Conference on Computer Vision (ICCV) (2021): 387-396.

**Questions:**

No additional questions at this time (beyond those alluded to in the weaknesses section).

**Limitations:**

Some limitations are indeed addressed throughout. However, (unless I have missed something, in which case I apologise!) I can only find the limitations of the small number of datasets used stated in the NeurIPS checklist. This needs to be stated explicitly in the main paper.

---

> ### Author Rebuttal · Authors · 2024-08-06
>
> **[W1] Feature importance comparisons**
>
> We did not perform an in-depth analysis of the feature importance extraction because it was not the main focus of our work. The primary goals of this work are twofold: first, to theoretically model the exploration of a Transformer's input space, and second, to implement this theoretical results using SiMEC, SiMExp, and input exploration techniques to be found in Section 3. These objectives are stated in lines 7-8, 12-13, 20-23, 319-323. However, the reviewer's proposal concerning the effect of ablation on the feature importance analysis is indeed interesting. We expect that ablations -- for example covering one patch of an picture with a black square in a image classification task -- do have an impact on the identification of important features. In this classification example, the mathematical model depicted in the first half of the work guarantees that modifying patches with high eigenvalues leads to a greater change in output compared to patches associated with low eigenvalues. On the other hand the pullback metric formula (5) suggests that if the network is very sensible to a change of a particular patch, then also the metric can change more. These two facts allude that the ablation or perturbation of important features may affect their eigenvalues more compared to features which are less important. Apart from these qualitative considerations, understanding exactly how ablation affects the importance of certain features requires both a theoretical and an empirical analysis. At this stage these are out of the scope, but it could be a building block for a future work.
>
> **[W2] Limited experimental results & improvements**
>
> We acknowledge the simplicity of our experimental methodology. In light of the aforementioned goals, our intention was not to introduce a new technique for feature importance, but rather to demonstrate the potential empirical applications of our approach and to discuss how our proposed tools can facilitate Explainable AI tasks. We selected the MNIST dataset for our experiments due to its simplicity and popularity, which we believe facilitates easier interpretation by the reader. The well-known differences between handwritten digits in MNIST make it straightforward to understand how changes in input data can alter outputs (e.g., adding a straight line to the digit '5' to make it resemble a '9'). We agree that TinyImageNET could have been an interesting dataset for our experiments; however, we prioritized clarity in our results over potentially more intriguing examples.
>
> As already stated, our main goal is not to convince the reader to adopt our method for assessing feature importance, but to show that our theory works correctly on the Transformers input space and for that reason produces an outcome that is competitive with State-of-the-Art techniques in the field of feature importance-based explanations. In particular, the two baselines were chosen with the goal of comparing our method with a well-established method specifically designed for Transformers (i.e. Attention Rollout) and a method that was recently proven to outperform many others, including Grad-CAM (see reference [6]).

---

> > ### Comment · Reviewer_KhQb · 2024-08-08
> > **Thanks to the authors**
> >
> > Thanks to the authors for their response.
> >
> > I appreciate the authors’ efforts to clarify the goals and claims of the paper in their first response.
> >
> > Unfortunately, I am still not convinced as it stands that the limited experimental results on MNIST sufficiently demonstrate the authors’ claim made in the rebuttal to `"demonstrate the potential empirical applications of our approach"` . I do not agree with the authors that adding additional results would have traded-off clarity (these can always be deferred to the supplementary material, for example).
> >
> > Furthermore, two additional reviewers have the same flavor of concerns—it would have been easy for the authors to conduct small experiments on any of the additional suggested datasets (such as CIFAR by **Reviewer-TeNK** or TinyImageNET as suggested in the initial review). In the absence of additional experiments, my rating currently remains the same.

---

### Official Review · Reviewer_b3eP · 2024-07-16

**Soundness:** 2
**Presentation:** 3
**Contribution:** 1
**Rating:** 3
**Confidence:** 4

**Summary:**

This work attempts to find the set of inputs that generate the same neural network predictions. To this end, the authors interpret the layers of the network as transformations of the input manifold. This interpretation is used to defined equivalence classes over the inputs and to define feature importance. Finally, the tools are used to identify equivalence classes for MNIST digits and for hate speech detection with BERT.

**Strengths:**

Section 2 does a thorough job of introducing a manifold interpretation to neural networks.  This introduction is then used to motivate multiple algorithms for finding equivalence classes of the inputs---or the setup of inputs that result in the same prediction---and identify features that are important.

**Weaknesses:**

The main contribution of this work is to introduce a tool for analyzing which set of inputs produce the same output. However, this is exactly the Fisher information matrix (with respect to the inputs) and has also been introduced in prior work (https://arxiv.org/abs/2104.13289). Could the authors clarify what the differences are and what the additional novelty. If the "local data matrix" introduced in https://arxiv.org/abs/2104.13289 is identical to the tools in this work, I think it severely diminishes the contributions of this work. Furthermore the experiments are extremely similar (such as Figure 1).

The second weakness is the limited number of experiments. The work does not show any quantitative results: Figure 1, Figure 2 and Figure 3 is just 1 example and is not indicative of why the tools are useful. The experiments in section 4 primarily discuss wall-clock time. It would significantly help if claims such as "(Line 266) we notice that the perturbation-based algorithm ends up producing monochrome ..." are substantiated quantitatively. Overall, the work doesn't provide novel tools and the experiments lack a novel usage of these tools and do not reveal any new insights.

**Questions:**

1. How does this work differ from https://arxiv.org/abs/2104.13289?
2. What is algorithm 4 (exploration) used for?
3. Many experimental details are lacking. It would help to add appendices clarifying what networks were used and how they were trained. Overall, the work feels incomplete in many respects
4. For feature importance-based explanations, there are many other methods like LIME/SHAP (and many more follow up papers) which the authors do not compare to. Why were they omitted?
5. Does this method scale to larger datasets, considering we have to compute outer-product of the gradients and compute the Eigenvalues of this matrix?

**Limitations:**

The authors address limitations of their work but it can be expanded upon. For example, the authors can discuss the time required to compute Eigenvalues, and other limitations such as not having any Eigenvalues to be 0 in Algorithms 3 / 4. Furthermore, their algorithm should work (in theory) for infinitesimal steps in the input manifold.

---

> ### Author Rebuttal · Authors · 2024-08-06
>
> We will address the questions in the same order as presented by the reviewer.
>
> **1. How does this work differ from https://arxiv.org/abs/2104.13289?**
>
> The two works present several differences, we enlist them down below.
> - The theoretical framework in our work can be applied for very general settings, there are no particular requirements on the loss function, while in the suggested work only the Kullback-Leibler divergence is considered.
> - We allow the employment of a large class of activation functions, while in https://arxiv.org/abs/2104.13289 only ReLu and softmax are considered.
> - We do not require that the matrix representing the pullback has constant rank: this is actually a consequence of the mild assumptions on our layers.
> - Our approach allows the study of the behaviour of \emph{fully trained} networks: of course one can apply our method also on partially trained ones.
> - On the contrary of https://arxiv.org/abs/2104.13289, we discuss the intensities of the eigenvalues, in order to employ them for feature importance.
> - The linked document exploits just the powerful, although simple, second order Taylor expansion of the loss, which hence must be at least belong to $\mathcal{C}^2(\Omega)$, being $\Omega$ the loss' domain. In our work, the theoretical foundations is based on proved theorems and general results: hence, our approach does not depend on the loss function nor on its differentiability.
>
> **2. What is algorithm 4 (exploration) used for?**
>
> Algorithm 4 outlines a specialized procedure for exploring the Transformer's input space, allowing for the selection of specific patches for exploration (see lines 189-195). This algorithm generates embeddings based on the exploration conducted over a set number of iterations. To make the changes between iterations perceptible to the human eye, a subsequent interpretation step is required. This is where Algorithm 5 comes in; it takes the embeddings produced by Algorithm 4 as its input. Consequently, when discussing the interpretation of the exploration process, we refer to Algorithm 5. As noted on line 3 of Algorithm 5, the algorithm explicitly requires the outputs from Algorithm 4 to function.
>
> **3. Many experimental details are lacking. It would help to add appendices clarifying what networks were used and how they were trained. Overall, the work feels incomplete in many respects.**
>
> The experiments utilize two Transformer networks: (1) a BERT model for hate speech detection, used without further training, as sourced from Hugging Face (huggingface.co/ctoraman/hate-speech-bert), as mentioned in Note 4; and (2) a Vision Transformer (ViT) model, consisting of 4 layers with 4 heads per layer, trained on the MNIST dataset using the Adam optimizer over 20 epochs, as described in Note 5. All the code used for modeling and training the networks is provided in the Supplementary Materials.
>
> **4. For feature importance-based explanations, there are many other methods like LIME/SHAP (and many more follow up papers) which the authors do not compare to. Why were they omitted?**
>
> Since feature importance-based explanations are not the primary focus of our work, we present them only as an application, or a collateral outcome, of our method, whose central goal is the exploration of the input space of Transformers. As a consequence, we simply show that our SiMEC-based feature importance explanations are competitive with State-of-the-Art methods commonly used in the context of Transformers. Techniques such as LIME and SHAP were judged relevant in the field of feature importance evaluation, but other techniques (i.e. Attention Rollout and the Relevancy method) were deemed more appropriate and up-to-date in the realm of Transformers.
>
> **5. Does this method scale to larger datasets, considering we have to compute outer-product of the gradients and compute the Eigenvalues of this matrix?**
>
> The key factors in estimating computational complexity and scalability are primarily related to the network architecture. As mentioned in lines 160-164, the most computationally intensive task is calculating the eigenvalues and eigenvectors, which has a complexity of $O(n^3)$, with $n$ the dimension of the square matrix $g^0_{p_k}$ (see reference 20), thus the embedding's dimension. Since all other operations have complexities of either $O(n)$ or $O(n^2)$, the overall complexity for both SIMEC and SIMExp is $O(n^3)$. Since the complexity depends on the embedding's dimension and not on the number of instances, the entire procedure scales linearly with respect to the number of instances processed in the experiments.

---

> > ### Comment · Reviewer_b3eP · 2024-08-08
> > **Thank you for the response**
> >
> > Thank you for the taking the time to respond to all the questions. However many of my concerns remain unresolved. I agree with the authors that there are some differences with prior work that uses the local Fisher/data matrix. However, the differences about the activation function or loss function seem of little relevance in the context of deep networks. Unless I am mistaken, the algorithm used in (https://arxiv.org/pdf/2104.13289) is identical to algorithm 1.
> >
> > As mentioned in the weakness section, the experiments are limited to small datasets and 3 of the 4 results are qualitative. The purpose of algorithm 4/5 is still not clear to me; I understand the steps of the algorithm, but I don't understand what goal it achieves and there are no results that show it can be used to derive new insights. The authors have introduced new tools / algorithms but I believe that there needs to be more thorough experimental validation.

---

> > > ### Author Response · Authors · 2024-08-09
> > >
> > > We thank the reviewer for their effort in replying to our rebuttal response. However, we still do not agree with the reviewer’s point of view about the novelty of both our mathematical framework and algorithms. We are confident in the distinctiveness of our work compared to https://arxiv.org/pdf/2104.13289. In addressing the reviewers' concerns about its novelty, we thoroughly examined both the mathematical foundations and the application to Transformer architecture. Our responses highlighted six key differences and clarified our methodological contributions, emphasizing the relevance of all presented algorithms. While there may be surface-level similarities, our algorithms are underpinned by unique mathematical proofs that ensure broader generalizability (without restrictions on loss functions and covering a wider range of activation functions), applicability to trained networks (and not only on partially trained ones), and the flexibility to operate without assuming a constant rank for the pullback metric. Based on our comparison, we believe there are substantial differences between the two works, and the divergence points that we highlighted provide strong evidences against the reviewer’s argumentation.

---

### Official Review · Reviewer_EZRw · 2024-07-19

**Soundness:** 3
**Presentation:** 2
**Contribution:** 2
**Rating:** 6
**Confidence:** 2

**Summary:**

The authors present a method for exploring equivalence classes in the input space of Transformer models using a solid mathematical theory. By analyzing the Jacobian of the model, the method reconstructs and navigates these classes, offering a powerful tool for understanding Transformer interpretations and enhancing explainability. The proposed method is expected to solve problems in Computer Vision and Natural Language Processing tasks.

**Strengths:**

I write both strength and weakness.

First, I must disclose that I have no prior study or background in both Transformers and manifolds. While I am conceptually aware of them, my knowledge is limited to that extent, and I lack confidence in reviewing the technical details. Therefore, please consider my review comments as feedback from a layperson in this field, focusing on the overall mathematical consistency and readability of the paper.

This paper describes mathematics in a clear and understandable manner that even a layperson like myself can grasp. Each definition and theorem is stated accurately, and I believe that the general concepts can be understood with basic knowledge.

I personally feel that the objective of this paper is not clearly conveyed. While the paper claims to contribute to explainability and sensitivity analysis through the analysis of input manifolds, the logic behind this was not clear to me in the Introduction and Preliminaries. Although the concepts of explainability and sensitivity analysis become clearer in the later chapters, it might be beneficial to provide a bit more explanation in the Introduction.

Additionally, it might be helpful to clearly define the equivalence class mathematically.

Since I am not familiar with the existing literature, I was unable to judge the novelty of this work.

**Weaknesses:**

See above.

**Questions:**

See above.

**Limitations:**

N/A.

---

> ### Author Rebuttal · Authors · 2024-08-06
>
> The primary goals of this work are twofold: first, to theoretically model the exploration of a Transformer's input space, and second, to implement this theoretical results using SiMEC, SiMExp, and input exploration techniques to be found in Section 3. These objectives are stated in lines 7-8, 12-13, 20-23, 319-323. In light of these main goals, we also discuss how our proposed tools can facilitate Explainable AI tasks. Our experiments are not intended to introduce a new technique for feature importance, but rather to demonstrate the potential empirical applications of our approach.

---

### Official Review · Reviewer_TeNK · 2024-07-28

**Soundness:** 3
**Presentation:** 2
**Contribution:** 2
**Rating:** 5
**Confidence:** 2

**Summary:**

This paper develops a novel theoretical framework grounded in Riemannian geometry for analyzing the input space of Transformer models, and introduce two algorithms, SiMEC and SiMExp, which facilitate the exploration and interpretation of equivalence classes within this input space. These methods offer new insights the internal mechanisms of Transformers, and provide new understanding of how these models perceive and process input data which can be very useful in the field of explainable AI.

**Strengths:**

1 novelty: This paper provide an innovative application of Riemannian geometry to analyze the input spaces of Transformer models, which is very novel in the area.

2 Theory: This paper establishes a solid mathematical theory on how Riemannian geometry is applied to Transformer models. Based on this theory, SiMEC and SiMExp are developed to explore the input spaces of Transformer models.

**Weaknesses:**

In experiment, the MNIST dataset is a little bit trivial, as the pixels of the background is essentially zero. It is nice to see the application of the proposed algorithm on natural images like CIFAR.

**Questions:**

See above

**Limitations:**

Yes

---

> ### Author Rebuttal · Authors · 2024-08-06
>
> We acknowledge the simplicity of our experimental methodology. However, the primary goals of this work are twofold: first, to theoretically model the exploration of a Transformer's input space, and second, to implement this theoretical results using SiMEC, SiMExp, and input exploration techniques to be found in Section 3. These objectives are stated in lines 7-8, 12-13, 20-23, 319-323. In light of these main goals, we also discuss how our proposed tools can facilitate Explainable AI tasks. Our experiments are not intended to introduce a new technique for feature importance, but rather to demonstrate the potential empirical applications of our approach. We selected the MNIST dataset for our experiments due to its simplicity and popularity, which we believe facilitates easier interpretation by the reader. The well-known differences between handwritten digits in MNIST make it straightforward to understand how changes in input data can alter outputs (e.g., adding a straight line to the digit '5' to make it resemble a '9').

---

### Decision · Program_Chairs · 2024-09-25

**Decision:**

Reject

**Comment:**

The paper presents a method for exploring the equivalence classes of inputs of transformer models. Reviewers praised the clarity of exposition and mathematical rigor of the paper, which presents a theoretical framework based on differential geometry that is nevertheless readable for non-experts. The most significant weaknesses mentioned by the reviewers are the lack of quantitative results (b3eP), lack of comparisons to existing feature importance methods (b3eP, KhQb) and the use of simple datasets like MNIST (TeNK).

In terms of ratings, we have a weak accept and accept (both with low confidence (with EZRw stating they are not expert on the topic), a confident reject and a low confidence borderline reject.

In my view, the theoretical content of the paper is solid and interesting, but the paper is not yet ready for publication. I would recommend that the authors focus on experimenting with more challenging datasets, since results on MNIST often don't transfer. It would also be good to establish quantitative metrics to evaluate the success of the method, and make sure to do a thorough review of the literature and compare to all relevant prior works.